# NEAR-OPTIMAL ALGORITHMS FOR GROUP DISTRIBUTIONALLY ROBUST OPTIMIZATION AND BEYOND

## ABSTRACT

Distributionally robust optimization (DRO) can improve the robustness and fairness of learning methods. In this paper, we devise stochastic algorithms for a class of DRO problems including group DRO, subpopulation fairness, and empirical conditional value at risk (CVaR) optimization. Our new algorithms achieve faster convergence rates than existing algorithms for multiple DRO settings. We also provide a new information-theoretic lower bound that implies our bounds are tight up to a log factor for group DRO. Empirically, too, our algorithms outperform known methods.

## 1 INTRODUCTION

Commonly, machine learning models are trained to optimize the average performance. However, such models may not perform equally well among all demographic subgroups due to a hidden bias in the training set or distribution shift in training and test phases (Hovy & Søgaard, 2015; Hashimoto et al., 2018; Martinez et al., 2021; Duchi & Namkoong, 2021). Biases in datasets are also directly related to fairness concerns in machine learning (Buolamwini & Gebru, 2018; Jurgens et al., 2017).

Recently, various algorithms based on distributionally robust optimization (DRO) have been proposed to address these problems (Hovy & Søgaard, 2015; Hashimoto et al., 2018; Hu et al., 2018; Oren et al., 2019; Williamson & Menon, 2019; Sagawa et al., 2020; Curi et al., 2020; Zhang et al., 2021; Martinez et al., 2021; Duchi & Namkoong, 2021). However, these algorithms are often highly tailored to each specific DRO formulation. Furthermore, it is often unclear whether these proposed algorithms are optimal in terms of the convergence rate. Are there a unified algorithmic methodology and a lower bound for these problems?

**Contributions.** In this paper, we study a general class of DRO problems, which includes group DRO (Hu et al., 2018; Oren et al., 2019; Sagawa et al., 2020), subpopulation fairness (Martinez et al., 2021), conditional value at risk (CVaR) optimization (Curi et al., 2020), and many others. Let $\Theta \subseteq \mathbb{R}^n$ be a convex set of model parameters and $\ell(\theta; z) : \Theta \to \mathbb{R}_+$ be a convex loss of the model with parameter $\theta$ with respect to data point $z$. The data point $z$ may be drawn from one out of $m$ distributions $P_1, \ldots, P_m$ which are accessible via a stochastic oracle that returns an i.i.d. sample $z \sim P_i$. Let $Q$ be a convex subset of the probability simplex in $\mathbb{R}^m$ that contains the uniform vector, i.e., $(1/m, \ldots, 1/m) \in Q$. In this paper, we conside the following DRO

$$\min_{\theta \in \Theta} \max_{q \in Q} \sum_{i=1}^{m} q_i \mathbb{E}_{z \sim P_i}[\ell(\theta; z)], \tag{1}$$

which we call *generalized group DRO*. If $Q$ are the probability simplex and scaled $k$-set polytope, we can recover group DRO (Sagawa et al., 2020) and subpopulation fairness (Martinez et al., 2021), respectively. Moreover, we formulate a new, more general fairness concept based on weighted rankings with $Q$ being a permutahedron, which includes these special cases; see Section 2 for details.

For generalized group DRO, we devise an efficient stochastic gradient algorithm. Furthermore, we show that it almost achieves the information-theoretic optimal convergence rate for group DRO up to a log factor. Our main technical contributions are as follows;

- We provide a generic stochastic gradient algorithm for generalized group DRO. By specializing it in the group DRO setting, we provide two algorithms (GDRO-EXP3 and GDRO-

Table 1: Summary of convergence results for group DRO. Here, $m$ denotes the number of groups, $n$ the dimension of $\theta$, $G$ the Lipschitz constant of loss function $\ell$, $D$ the diameter of feasible set $\Theta$, $M$ the range of loss function $\ell$, and $T$ the number of calls to stochastic oracle. The convergence of (Sagawa et al., 2020) and Theorem 2 are with respect to $\mathbb{E}[\epsilon_T]$ while the convergence of Theorem 3 is weaker with respect to $\mathbb{E}[\epsilon_T(q^*)]$ for saddle point $(\theta^*, q^*)$

| reference | convergence rate $\mathbb{E}[\epsilon_T]^{\dagger}$ or $\mathbb{E}[\epsilon_T(q^*)]^{\ddagger}$ | iteration complexity | lower bound |
|---|---|---|---|
| (Sagawa et al., 2020) | $O\left(m\sqrt{\frac{G^2D^2+M^2\log m}{T}}\right)^{\dagger}$ | $O(m+n)$ + proj. onto $\Theta$ | |
| **Ours (Theorem 2)** | $O\left(\sqrt{\frac{G^2D^2+M^2m\log m}{T}}\right)^{\dagger}$ | $O(m+n)$ + proj. onto $\Theta$ | $\Omega\left(\sqrt{\frac{G^2D^2+M^2m}{T}}\right)$ **(Theorem 5)** |
| **Ours (Theorem 3)** | $O\left(\sqrt{\frac{G^2D^2+M^2m}{T}}\right)^{\ddagger}$ | $O(m+n)$ + proj. onto $\Theta$ + solving scalar equation | |

TINF) that improve the rate of Sagawa et al. (2020) by a factor of $\Omega(\sqrt{m})$ with the almost same complexity per iteration; see Table 1. Furthermore, our generic algorithm can be specialized to improve the convergence rate of Curi et al. (2020) for subpopulation fairness (a.k.a. empirical CVaR optimization). Finally, we show that our algorithm runs efficiently if $Q$ is a permutahedron, which includes all aforementioned subclasses.

- We prove an almost matching information-theoretic lower bound for the convergence rate of group DRO. This implies that no algorithm can improve the convergence rate of GDRO-EXP3 (up to a constant factor). To the best of our knowledge, this is the first information-theoretic lower bound for group DRO.

- Our experiments on real-world and synthetic datasets show that our algorithms also empirically outperform the known algorithm, supporting our theoretical analysis. Although our convergence analysis only holds for the convex regime, our proposed algorithms outperform even in the deep learning regime.

## 1.1 OUR TECHNIQUES

**Algorithms.** The core idea of our algorithms is *stochastic no-regret dynamics* (Hazan, 2016). We regard DRO equation 1 as a two-player zero-sum game between a player who picks $\theta \in \Theta$ and another player who picks $q \in Q$. The two players iteratively update their solution using online learning algorithms; in particular, we will use online gradient descent (OGD) (Zinkevich, 2003) and online mirror descent (OMD) (Cesa-Bianchi & Lugosi, 2006) for the $\theta$-player and $q$-player, respectively. In addition, we need to estimate gradients for both players, since the objective function of generalized group DRO is stochastic and we cannot obtain exact gradients.

The convergence rate of stochastic no-regret dynamics depends on the expected regret of OGD and OMD. To obtain a near-optimal convergence rate, we must carefully choose the regularizer in OMD as well as gradient estimators, exploiting the structure of generalized group DRO. In particular, we need to balance the variance of gradient estimators and the diameter terms in *both* OGD and OMD. This is the most challenging part of the algorithm design. Inspired by adversarial multi-armed bandit algorithms, we design gradient estimators for no-regret dynamics of OGD and OMD in generalized group DRO. Indeed, our algorithms for group DRO (GDRO-EXP3 and GDRO-TINF) are based on adversarial multi-armed bandit algorithms, EXP3 (Auer et al., 2003) and Tsallis-INF (Zimmert & Seldin, 2021), respectively, hence the name. Although each building block (OGD, OMD, and gradient estimators) is fairly known in the literature, we need to put them together in the right combination to obtain the correct rate.

**Lower bound.** For the lower bound, we carefully design a family of group DRO instances for which any algorithm requires a certain number of queries to achieve a good objective value. To bound the number of queries, we use information-theoretic tools such as Le Cam's lemma and bound the Kullback-Leibler divergence between Bernoulli distributions. Such tools are also used at the heart of lower bounds for stochastic convex optimization (Agarwal et al., 2012) and adversarial multi-armed

bandits (Auer et al., 2003), but the connection to those settings is much more subtle here, and our construction is specifically designed for group DRO-type problems.

## 1.2 RELATED WORK

DRO is a wide field ranging from robust optimization to machine learning and statistics (Goh & Sim, 2010; Bertsimas et al., 2018), whose original idea dates back to Scarf (1958). Popular choices of the uncertainty set in DRO include balls around an empirical distribution in Wasserstein distance (Esfahani & Kuhn, 2018; Blanchet et al., 2019), $f$-divergence (Namkoong & Duchi, 2016; Duchi & Namkoong, 2021), $\chi^2$-divergence (Staib et al., 2019), and maximum mean discrepancy (Staib & Jegelka, 2019; Kirschner et al., 2020).

DRO algorithms have been mainly studied for the offline setting, i.e., algorithms can access all data points of the empirical distribution. Note that generalized group DRO is not offline because the group distributions are given by the stochastic oracles. Namkoong & Duchi (2016) proposed stochastic gradient algorithms for offline DRO with $f$-divergence uncertainty sets. Curi et al. (2020) used no-regret dynamics for empirical CVaR minimization. Their algorithm invokes sampling from $k$-DPP in each iteration, which is more computationally demanding than our algorithm. Furthermore, our algorithm gets rid of an $O(\log m)$ factor in the convergence rate using the Tsallis entropy regularizer; see Theorem 4. Qi et al. (2021); Jin et al. (2021) devised stochastic gradient algorithms for several DRO with non-convex losses.

Agarwal et al. (2012) gave a lower bound for stochastic convex optimization, which is a special case of generalized group DRO with only one distribution. Recently, Carmon et al. (2021) showed a lower bound for minimax problem $\min_x \max_{i=1}^m f_i(x)$ for non-stochastic Lipschitz convex $f_i$. Our lower bound deals with the stochastic functions, so this result does not apply.

In this paper, we assume that the group information is given in advance. However, the group information might not be easy to define in practice. Bao et al. (2021) propose a simple method to define groups for classification problems based on mistakes of models in the training phase. Their method often generates group DRO instances with large $m$. Our algorithms are more efficient for such group DRO thanks to the better dependence on $m$ in the convergence rate.

No-regret dynamics is a well-studied method for solving two-player zero-sum games (Cesa-Bianchi & Lugosi, 2006). For non-stochastic convex-concave games, one can achieve $O(1/T)$ convergence via predictable sequences (Rakhlin & Sridharan, 2013). This result does not apply to our setting because our DRO is a stochastic game.

**Notations.** Throughout the paper, $m$ denotes the number of distributions (groups) and $n$ denotes the dimension of a variable $\theta$. For a positive integer $m$, we write $[m] := \{1, \dots, m\}$. The orthogonal projection onto set $\Theta$ is denoted by $\mathrm{proj}_\Theta$. The $i$th standard unit vector is denoted by $\mathbf{e}_i$ and the all-one vector is denoted by $\mathbf{1}$. The probability simplex in $\mathbb{R}^m$ is denoted by $\Delta_m$.

## 2 EXAMPLES CONTAINED IN GENERALIZED GROUP DRO

In this section, we show how several DRO formulations in the literature can be phrased in generalized group DRO equation 1. In addition, we propose a novel fairness constraint based on weighted rankings using generalized group DRO.

**Group DRO.** When $Q$ equals the probablility simplex, we obtain original group DRO (Hu et al., 2018; Oren et al., 2019; Sagawa et al., 2020):

$$\min_{\theta \in \Theta} \max_{i=1}^m \; \mathbb{E}_{z \sim P_i}[\ell(\theta; z)]. \tag{2}$$

That is, group DRO aims to minimize the expected loss in the worst group, thereby ensuring better performance across all groups.

**Empirical CVaR, Subpopulation fairness, Average top-$k$ worst group loss.** Group DRO may yield overly pessimistic solutions. For instance, the groups might be automatically generated by

other algorithms (such as one in Bao et al. (2021)) and there might exist a few "outlier" groups that make the group DRO objective trivial.

For such a case, we can restrict $Q$ to a small subset of the probability simplex so that the solution cannot put large weights on a few outlier groups. Especially, let

$$Q = \left\{ q \in \Delta_m : 0 \leq q_i \leq \frac{1}{pm} \right\}$$

for some parameter $p \in (0, 1)$, i.e., $Q$ is a scaled $k$-set polytope. The intuition behind the choice of $Q$ is that, by limiting the largest entry of $q$ to $1/pm$, DRO would optimize the expected loss over the worst $p$-fraction subgroups of $m$ groups. Therefore, if the fraction of outlier groups is sufficiently small compared to $p$, then $p$-fraction subgroups must contain "inlier" groups as well. Therefore, it is likely that DRO with $Q$ finds solutions more robust than group DRO.

When $P_i$ is the Dirac measure of data $z_i$, then the resulting DRO is empirical CVaR optimization (Curi et al., 2020). In the fairness context, the same problem is called subpopulation fairness (Williamson & Menon, 2019; Martinez et al., 2021; Duchi & Namkoong, 2021).

If $p = k/m$ for some positive integer $k$, the resulting DRO is the average top-$k$ worst group loss (Zhang et al., 2021):

$$\min_{\theta \in \Theta} \frac{1}{k} \sum_{i=1}^{k} L_i^{\downarrow}(\theta),$$

where $L_i^{\downarrow}(\theta)$ denotes the the $i$th largest population group loss of $\theta$. More precisely, let $L_i(\theta) = \mathbb{E}_{z \sim P_i}[\ell(\theta; z)]$ for $i \in [m]$ and sort them in the non-increasing order: $L_1^{\downarrow}(\theta) \geq \cdots \geq L_m^{\downarrow}(\theta)$.

**Weighted ranking of group losses.** The aforementioned DRO formulations are special cases of the following DRO, which we call the *weighted ranking of group losses*. Let $\alpha \in \Delta^m$ be a fixed vector with non-increasing entries. Let $Q$ be the permutahedron of $\alpha$, the convex hull of $(\alpha_{\sigma(1)}, \ldots, \alpha_{\sigma(m)})$ for all permutations $\sigma$ of $[m]$. Then, the resulting DRO is

$$\min_{\theta \in \Theta} \sum_{i=1}^{m} \alpha_i L_i^{\downarrow}(\theta).$$

Group DRO corresponds to $\alpha = (1, 0, \ldots, 0)$ and the average top-$k$ worst group losses corresponds to $\alpha = \underbrace{(1/k, \ldots, 1/k}_{k \text{ times}}, 0, \ldots, 0)$. Another example that is contained in none of the above examples is *lexicographic minimax fairness* (Diana et al., 2021). The goal of lexicographical minimax fairness is to find $\theta \in \Theta$ such that the sequence $(L_1^{\downarrow}(\theta), \ldots, L_m^{\downarrow}(\theta))$ is lexicographically minimum. This corresponds to $\alpha$ with sufficiently varied entries, i.e., $\alpha_1 \gg \alpha_2 \gg \cdots \gg \alpha_m$.

## 3 ALGORITHMS

In this section, we describe our algorithms. First, we present a generic algorithm for generalized group DRO equation 1 and provide a unified convergence analysis in Section 3.1. Then, we specialize it into two concrete algorithms for group DRO equation 2 in Section 3.2. We sketch algorithms for the average of top-$k$ group losses and weighted ranking of group loss in Section 3.3.

### 3.1 ALGORITHM FOR THE GENERAL CASE

We present our algorithm for generalized group DRO equation 1. At a high level, our algorithm can be regarded as stochastic no-regret dynamics. Let us denote $L(\theta, q) := \sum_{i=1}^{m} q_i \mathbb{E}_{z \sim P_i}[\ell(\theta; z)]$. Imagine that the $\theta$-player and $q$-player run online algorithms $\mathcal{A}_\theta$ and $\mathcal{A}_q$, respectively, to solve the minimax problem $\min_{\theta \in \Theta} \max_{q \in Q} L(\theta, q)$. That is, for $t = 1, \ldots, T$,

- $\theta_t \in \Theta$ and $q_t \in Q$ are determined by $\mathcal{A}_\theta$ and $\mathcal{A}_q$, respectively.
- Both players feed gradient estimators $\hat{\nabla}_{\theta,t}$ and $\hat{\nabla}_{q,t}$ to $\mathcal{A}_\theta$ and $\mathcal{A}_q$, respectively. Here, $\mathbb{E}[\hat{\nabla}_{\theta,t}] = \nabla_\theta L(\theta_t, q_t)$ and $\mathbb{E}[\hat{\nabla}_{q,t}] = \nabla_q L(\theta_t, q_t)$.

Let $\theta^*$ be an optimal solution. Let

$$\epsilon_T := \max_{q \in Q} L(\bar{\theta}_{1:T}, q) - \max_{q \in Q} L(\theta^*, q).$$

be the optimality gap of the averaged iterate $\bar{\theta}_{1:T} = \frac{1}{T} \sum_{t=1}^{T} \theta_t$. In o

We can bound the expected convergence rate $\mathbb{E}[\epsilon_T]$ via regrets $R_\theta$ and $R_q$ of these online algorithms (see Appendix A for a formal definition), i.e.,

$$\mathbb{E}[\epsilon_T] \leq \frac{\mathbb{E}[R_\theta(T)] + \mathbb{E}[R_q(T; \theta^*)]}{T}. \tag{3}$$

We can obtain hence the convergence rate of the above algorithms by investigating the expected regret bounds of these online algorithms.

We also use the following weaker notion of convergence. For any *fixed* $q^* \in Q$, let

$$\epsilon_T(q^*) := L(\bar{\theta}_{1:T}, q^*) - L(\theta^*, q^*),$$

be the gap of $\bar{\theta}_{1:T}$ with respect to $\theta^*$. By definition, $\mathbb{E}[\epsilon_T(q^*)] \leq \mathbb{E}[\epsilon_T]$ for any $q^* \in Q$. If $\mathbb{E}[\epsilon_T(q^*)] \leq \epsilon$, then $\bar{\theta}_{1:T}$ is an $\epsilon$-approximate saddle point, i.e.,

$$L(\theta^*, q^*) \leq \mathbb{E}[L(\bar{\theta}_{1:T}, q^*)] \leq L(\theta^*, q^*) + \epsilon.$$

To get a concrete algorithm, we must specify the online algorithms $\mathcal{A}_\theta, \mathcal{A}_q$ as well as the gradient estimators $\hat{\nabla}_{\theta,t}, \hat{\nabla}_{q,t}$. We use OGD and OMD as $\mathcal{A}_\theta$ and $\mathcal{A}_q$, respectively. We construct the gradient estimators by sampling $i_t \sim q_t$ and $z \sim P_{i_t}$ and setting $\hat{\nabla}_{\theta,t} = \nabla_\theta \ell(\theta_t; z)$ and $\hat{\nabla}_{q,t} = \frac{\ell(\theta_t; z)}{q_{t, i_t}} \mathbf{e}_{i_t}$. This leads to Algorithm 1. There, $\Psi : Q \rightarrow \mathbb{R}$ denotes the regularizer of OMD and $\eta_{\theta,t}$ and $\eta_q$ denote the step sizes of OGD and OMD, respectively. [1] It turns out that this combination of online algorithms and gradient estimators yields the best convergence rate (for group DRO) because the expected regrets of both players are optimal.

---

**Algorithm 1** Algorithm for generalized group DRO equation 1

---

**Require:** initial solution $\theta_1 \in \Theta$, number of iterations $T$, step sizes $\eta_{\theta,t} > 0$ ($t \in [T]$), $\eta_q > 0$, and a strictly convex function $\Psi : Q \rightarrow \mathbb{R}$.
1: Let $q_1 = (1/m, \ldots, 1/m)$.
2: **for** $t = 1, \ldots, T$ **do**
3:     Sample $i_t \sim q_t$.
4:     Call the stochastic oracle to obtain $z \sim P_{i_t}$.
5:     $\theta_{t+1} \leftarrow \text{proj}_\Theta(\theta_t - \eta_{\theta,t} \nabla_\theta \ell(\theta_t; z))$
6:     $\nabla\Psi(\tilde{q}_{t+1}) \leftarrow \nabla\Psi(q_t) - \frac{\eta_q}{q_{t,i_t}} \ell(\theta_t; z) \mathbf{e}_{i_t}$;   $q_{t+1} \leftarrow \arg\min_{q \in Q} D_\Psi(q, \tilde{q}_{t+1})$, where $D_\Psi(x, y) = \Psi(x) - \Psi(y) - \nabla\Psi(x)^\top(y - x)$ is the Bregman divergence with respect to $\Psi$.
7: **return** $\frac{1}{T} \sum_{t=1}^{T} \theta_t$.

---

We now analyze the convergence rate of Algorithm 1. We make the following standard assumptions.

**Assumption 1.** *The loss function $\ell(\theta; z)$ is continuously differentiable and $G$-Lipchitz in $\theta$, and has range $[0, M]$ for all $z$. The Euclidean diameter of the feasible region $\Theta$ is at most $D$.*

The following theorem follows from plugging regret bounds of OGD and OGD, and the construction of the gradient estimators into equation 3.

**Theorem 1.** *If $\eta_{\theta,t}$ is nonincreasing, Algorithm 1 achieves the expected convergence rate*

$$\mathbb{E}[\epsilon_T(q^*)] \leq \frac{1}{T} \left( \frac{G^2}{2} \sum_{t=1}^{T} \eta_{\theta,t} + \frac{D^2}{2\eta_{\theta,T}} + \frac{M^2}{2} \eta_q \sum_{t=1}^{T} \mathbb{E}_{i_t} \left[ \frac{(\nabla^2 \Psi(q_t))_{i_t,i_t}^{-1}}{q_{t,i_t}^2} \right] + \frac{D_\Psi(q^*, \mathbf{1}/m)}{\eta_q} \right).$$

*for any $q^* \in Q$.*

A formal proof can be found in Appendix B. We will see how specific choices of the regularizer $\Psi$ yield various algorithms and convergence rates for group DRO and others in the next subsections. A few remarks on the regularizers, step sizes, and projection step are in order.

---

[1] We make a standard assumption that the regularizer $\Psi$ is differentiable and strictly convex, and satisfies $\|\nabla\Psi(x)\| \rightarrow +\infty$ as $x$ tends to the boundary of $Q$.

**Regularizer.** Although Algorithm 1 works with general $\Psi$, we can choose a specific regularizer for $Q$ appearing in applications, e.g, the probability simplex, scaled $k$-set polytope, or a permutahedron. In the next subsections, we show that the entropy regularizer $\Psi(x) = \sum_i (x_i \log x_i - x_i)$ and Tsallis entropy regularizer $\Psi(x) = 2(1 - \sum_i \sqrt{x_i})$ yield efficient algorithms with improved convergence rates for these cases.

**Step sizes.** The theorem includes decreasing step sizes such as $\eta_{\theta,t} = \frac{D}{mG\sqrt{t}}$ in addition to fixed step sizes. Decreasing step sizes have the advantage that we do not require the knowledge of $T$ at the beginning of the algorithm but come at the cost of an extra constant factor in the expected convergence rate. Since both step size policies give the asymptotically same convergence rate, we describe only fixed step sizes in the theorems in the next subsections. In practice, decreasing step sizes stabilize the algorithm and often outperform fixed step sizes.

**Projection step.** In general, the Bregman projection $\arg\min_{q \in Q} D_\Psi(q, \tilde{q}_{t+1})$ is convex, but may be costly to compute. For the applications described in Section 2, $Q$ is a permutahedron. In this case, it is known that the Bregman projection with respect to the entropy and Tsallis entropy regularizers can be done in $O(m \log m)$ time (Lim & Wright, 2016). If $Q$ is the probability simplex, we even have a closed form for the Bregman projection.

## 3.2 Algorithms for Group DRO

We now describe two concrete algorithms for group DRO equation 2.

**GDRO-EXP3P.** The first algorithm is obtained by using the EXP3P algorithm for the $q$-player algorithm. The resulting algorithm, GDRO-EXP3P, is shown in Algorithm 2. The update is in a closed formula and its complexity is $O(m+n)$ time. The convergence rate follows from Theorem 1.

---

**Algorithm 2** GDRO-EXP3P

**Require:** initial solution $\theta_1 \in \Theta$, number of iterations $T$, and step sizes $\eta_{\theta,t} > 0$ ($t \in [T]$), $\eta_q > 0$, $\beta, \gamma > 0$.
1: Let $q_1 = (1/m, \ldots, 1/m)$.
2: **for** $t = 1, \ldots, T$ **do**
3:     Sample $i_t \sim q_t$.
4:     Call the stochastic oracle to obtain $z \sim P_{i_t}$.
5:     $\theta_{t+1} \leftarrow \text{proj}_\Theta(\theta_t - \eta_{\theta,t} \nabla_\theta \ell(\theta_t; z))$
6:     Let $\tilde{g}_t := \frac{-\ell(\theta_t; z)\mathbf{e}_{i_t} + \beta \mathbf{1}}{q_{t,i_t}}$, $G_t := \sum_{\tau=1}^{t} \tilde{g}_t$, and $Z := \sum_{i \in [m]} \exp(\eta G_{t,i})$.
7:     $q_{t+1} \leftarrow (1 - \gamma)\frac{\exp(\eta G_t)}{Z} + \frac{\gamma \mathbf{1}}{m}$.
8: **return** $\frac{1}{T}\sum_{t=1}^{T} \theta_t$.

---

**Theorem 2.** *If $\eta_{\theta,t}$ is nonincreasing,* GDRO-EXP3P *(Algorithm 2) achieves*

$$\mathbb{E}[\epsilon_T] \leq \frac{1}{T}\left(\frac{G^2}{2}\sum_{t=1}^{T}\eta_{\theta,t} + \frac{D^2}{2\eta_{\theta,T}} + \frac{mM^2}{2}\eta_q T + \frac{\log m}{\eta_q}\right). \tag{4}$$

*For $\eta_{\theta,t} = \frac{D}{G\sqrt{T}}$ and $\eta_q = \sqrt{\frac{2\log m}{mM^2 T}}$, we obtain*

$$\mathbb{E}[\epsilon_T] \leq \sqrt{2}\frac{\sqrt{G^2 D^2 + 2M^2 m \log m}}{\sqrt{T}}.$$

**Comparison to Sagawa et al. (2020).** Our algorithm improve the convergence rate of Sagawa et al. (2020) by a factor of $O(\sqrt{m})$; see Table 1. The reason lies in the choice of gradient estimator. All algorithms are stochastic no-regret dynamics. As outlined above, their convergence hence can be bounded by the regrets of the players, which depend on the variance of the local norm of the gradient estimators. Their strategy is based on uniform sampling that yields a variance of $O(m)$ for both players, whereas our bound is $O(\sqrt{m})$ thanks to the gradient estimators tailored to the regularizer of OMD. More details may be found in Appendix D.

**GRDO-TINF.** The second algorithm is given by using the Tsallis entropy regularizer for the $q$-player algorithm. The update of $q_t$ is now

$$\tilde{q}_{t+1} = q_t \left( \mathbf{1} - \frac{\eta_q \sqrt{q_t}}{q_{t,i_t}} \ell(\theta_t; z) \mathbf{e}_{i_t} \right)^{-2},$$

$$q_{t+1} := \left( \frac{1}{\sqrt{\tilde{q}_{t+1}}} - \alpha \mathbf{1} \right)^{-2},$$

where the multiplication, square-root, and power operations are entry-wise and $\alpha \in \mathbb{R}$ is the unique solution of equation $\sum_{i=1}^m \left( 1/\sqrt{\tilde{q}_{t+1,i}} - \alpha \right)^{-2} = 1$. The solution $\alpha$ can be computed via the Newton method. Practically, one can use $\alpha$ in the previous iteration to warm start the Newton method. In each iteration, the algorithm performs a single orthogonal projection onto $\Theta$, the Newton method for finding $\alpha$, and $O(m+n)$ operations to update $\theta_t, q_t$. The pseudocode is given in Algorithm 3.

---

**Algorithm 3** GDRO-TINF

---

**Require:** initial solution $\theta_1 \in \Theta$, number of iterations $T$, and step sizes $\eta_{\theta,t} > 0$ ($t \in [T]$), $\eta_q > 0$.
1: Let $q_t = (1/m, \ldots, 1/m)$.
2: **for** $t = 1, \ldots, T$ **do**
3:      Sample $i_t \sim q_t$.
4:      Call the stochastic oracle to obtain $z \sim P_{i_t}$.
5:      $\theta_{t+1} \leftarrow \mathrm{proj}_\Theta (\theta_t - \eta_{\theta,t} \nabla_\theta \ell(\theta_t; z))$
6:      $\tilde{q}_{t+1} \leftarrow q_t \left( \mathbf{1} - \frac{\eta_q \sqrt{q_t}}{q_{t,i_t}} \ell(\theta_t; z) \mathbf{e}_{i_t} \right)^{-2}$
7:      Compute $\alpha \in \mathbb{R}$ such that $\sum_{i=1}^m \left( 1/\sqrt{\tilde{q}_{t+1,i}} - \alpha \right)^{-2} = 1$.
8:      $q_{t+1} \leftarrow \left( \tilde{q}_{t+1}^{-1/2} - \alpha \mathbf{1} \right)^{-2}$
9: **return** $\frac{1}{T} \sum_{t=1}^T \theta_t$.

---

From Theorem 1, we obtain the following convergence rate.

**Theorem 3.** *If $\eta_{\theta,t}$ is nonincreasing,* GDRO-TINF *(Algorithm 3) achieves*

$$\mathbb{E}[\epsilon_T(q^*)] \leq \frac{1}{T} \left( \frac{G^2}{2} \sum_{t=1}^T \eta_{\theta,t} + \frac{D^2}{2\eta_{\theta,T}} + \sqrt{m} M^2 \eta_q T + \frac{\sqrt{m}}{\eta_q} \right) \tag{5}$$

*for any saddle point $(\theta^*, q^*)$, $q^* \in Q$. For $\eta_{\theta,t} = \frac{D}{G\sqrt{T}}$ and $\eta_q = \frac{1}{M\sqrt{T}}$, we obtain*

$$\mathbb{E}[\epsilon_T(q^*)] \leq \sqrt{2} \frac{\sqrt{G^2 D^2 + 4M^2 m}}{\sqrt{T}}.$$

## 3.3 Algorithm for weighted ranking of group losses

We now consider a more general case that $Q$ is a permutahedron. Applying Algorithm 1 with the Tsallis entropy regularizer, we obtain the following result.

**Theorem 4.** *If $\eta_{\theta,t}$ is nonincreasing and $Q$ is a permutahedron, Algorithm 1 with the Tsallis entropy regularizer achieves the convergence to an approximate-saddle point as Theorem 3. Furthermore, the iteration complexity is $O(m \log m + n)$.*

This implies a convergence rate of $O(\sqrt{\frac{G^2 D^2 + M^2 m}{T}})$ for empirical CVaR optimization, which improves $O(\sqrt{\frac{G^2 D^2 + M^2 m \log m}{T}})$ convergence by Curi et al. (2020). Furthermore, their iteration complexity is $O(m^3)$ due to the $k$-DPP sampling step, so our algorithm is even faster in terms of iteration complexity.

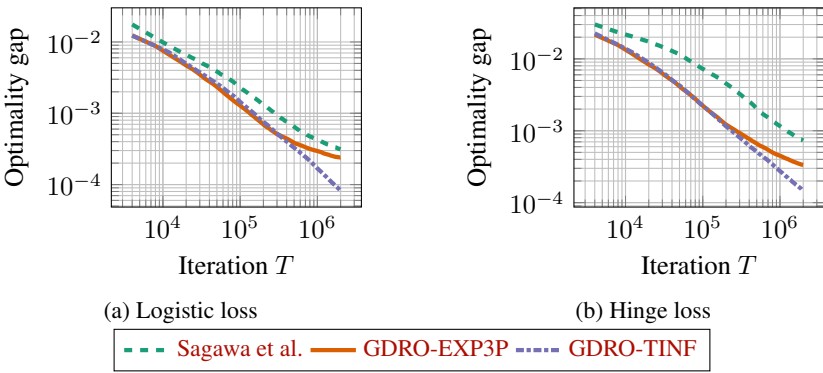

Figure 1: Results on Adult dataset for convex losses. Both axes are log-scale.

## 4 LOWER BOUND FOR GROUP DRO

Theorem 2 states that we can find an $\epsilon$-optimal solution for group DRO in $O(\frac{G^2D^2+M^2m\log m}{\epsilon^2})$ calls to stochastic oracles. Next, we show that this query complexity is almost information-theoretically optimal.

Let $\mathcal{L}$ be a class of convex $G$-Lipschitz loss functions $\ell : \Theta \to [0, M]$. Given a loss function $\ell \in \mathcal{L}$, and an $m$-set $\mathcal{P} = \{P_1, \ldots, P_m\}$ of distributions, denote the optimality gap of $\theta \in \Theta$ by

$$R(\theta, \ell, \mathcal{P}) = \max_{P \in \mathcal{P}} \mathbb{E}_{z \sim P}[\ell(\theta; z)] - \min_{\theta^* \in \Theta} \max_{P \in \mathcal{P}} \mathbb{E}_{z \sim P}[\ell(\theta^*; z)].$$

Let $\mathcal{A}_T$ be the set of algorithms that outputs $\hat{\theta} \in \Theta$ making $T$ queries to the stochastic oracle.

**Theorem 5** (Lower bound for group DRO)**.**

$$\inf_{\hat{\theta} \in \mathcal{A}_T} \sup_{\ell \in \mathcal{L}, \Theta, \mathcal{P}} \mathbb{E}_{\mathcal{P}}[R(\hat{\theta}, \ell, \mathcal{P})] \geq \Omega\left(\max\left\{\frac{GD}{\sqrt{T}}, M\sqrt{\frac{m}{T}}\right\}\right),$$

where $\Theta$ runs over convex sets with diameter $D$ and $\mathcal{P}$ over $m$-sets of distributions, and $\mathbb{E}_{\mathcal{P}}$ denotes the expectation over outcomes of the stochastic oracle in $\mathcal{P}$.

As $\sqrt{x+y} \leq \sqrt{x} + \sqrt{y} \leq \sqrt{2(x+y)}$ for $x, y \geq 0$, this theorem immediately implies that the minimax convergence rate is $\Omega\left(\sqrt{\frac{G^2D^2+M^2m}{T}}\right)$, which equals the convergence rate achieved by Algorithm 3 up to a constant factor.

**Proof Outline.** It suffices to show two lower bounds $\frac{GD}{\sqrt{T}}$ and $M\sqrt{\frac{m}{T}}$ independently. The former is a well-known lower bound for stochastic convex optimization (Agarwal et al., 2012). To illustrate the latter, we take an algorithmic dependent point of view via the Le cam's method. For any algorithm in $\mathcal{A}_T$, we need to construct instances $\mathcal{P}_0, \mathcal{P}_1$ such that the total variation distance between the distributions over the query outcomes (they depend on both the behavior of the algorithm and the instance) with respect to $\mathcal{P}_0$ and $\mathcal{P}_1$ is small. On the other hand, the objective function of the two instances must be well-separated, i.e., any fixed $\theta$ is $\delta$ sub-optimal for either $\mathcal{P}_0$ or $\mathcal{P}_1$. So, any algorithm that solves group DRO up to error $\delta$ needs to distinguish two instances $\mathcal{P}_0$ and $\mathcal{P}_1$. This implies a query lower bound because the total variation distance of the outcome distributions of these instances is small. The challenge is how to construct such instances for the regime of small dimensions of $\theta$, e.g, $n = 1$. To this end, we carefully construct linear functions for $m$ groups using opposite slopes. Then, based on the behavior of the algorithm, we tweak the noise bias in one of the groups with a positive slope, in a way that any fixed $\theta$ is $\Theta(\delta)$ sub-optimal for one of these instances. For the detailed proof, see Appendix C.

## 5 EXPERIMENTS

In this section, we compare our algorithms with baseline algorithms using real-world datasets in the group DRO setting for both convex and deep learning regimes. The additional detail of experiments

as well as an additional experiment are provided in Appendix E. The experiment codes are available in Supplementary materials.

## 5.1 EXPERIMENT IN THE CONVEX REGIME

First, we validate our convergence analysis for the convex regime. The experiment setup is adopted from Namkoong & Duchi (2016).

**Dataset.** We use Adult dataset (Dua & Graff, 2017), which consists of age, gender, race, educational background, and many other attributes of $48,842$ individuals from the US census. The task is to predict whether the person's income is greater than $50,000$ USD or not. We set up 6 groups based on the race and gender attributes: each group corresponds to a combination of $\{\text{black}, \text{white}, \text{others}\} \times \{\text{female}, \text{male}\}$. Converting the categorical features to dummy variables, we obtain a 101-dimensional feature vector $a \in \mathbb{R}^n$ ($n = 101$) for each individual. We train the linear model with the logistic loss and hinge loss functions. The group-DRO objective is the worst empirical loss over the 6 groups $\max_{i=1}^{6} \frac{1}{|I_i|} \sum_{(a,b) \in I_i} \ell(\theta; a, b)$,: where $I_i$ is the set of data points in the $i$th group. The feasible region is the Euclidean ball of radius $D = 10$.

### 5.1.1 ALGORITHMS

We implemented GDRO-EXP3P, GDRO-TINF, and the algorithm in (Sagawa et al., 2020) in Python. We ran our algorithms for $T = 2{,}000{,}000$ iterations.

**Step sizes.** The choice of step sizes is crucial to the practical performance of first-order methods. We found that the decreasing step size $\eta_{\theta,t} \sim 1/\sqrt{t}$ for $\theta_t$ and the fixed step size $\eta_q \sim 1/\sqrt{T}$ for $q_t$ gave the best results. More precisely, we set $\eta_{\theta,t} = \frac{C_\theta D}{\sqrt{t}}$ ($t \in [T]$) and $\eta_q = C_q \sqrt{\frac{\log m}{mT}}$, where $C_\theta \in [0.1, 5.0]$ and $C_q \in [0.1, 3.0]$ are hyper-parameters tuned for each algorithm. We used the best hyper-parameter found by Optuna (Akiba et al., 2019) for the shown results.

**Mini-batch and Initialization.** The use of mini-batch often improves the stability of stochastic gradient algorithms. In our experiments, we used mini-batches of size 10 to evaluate stochastic gradients. Neither the objective values of outputs nor the stability was improved with larger mini-batch sizes. The group DRO objective is evaluated using the entire dataset. Further, we initialized the algorithms with $\theta_1 = \mathbf{0}$.

### 5.1.2 RESULTS

In Figure 1, we plot the optimality gap of the averaged iterate $\frac{1}{T} \sum_{t=1}^{T} \theta_t$ against the number of iteration $T$. We observe that all the algorithms converge with a rate roughly $T^{-0.5}$ for both loss functions, consistent with our convergence bound. Furthermore, our algorithms (GDRO-EXP3P and GDRO-TINF) achieve faster convergence compared to the algorithm by Sagawa et al. (2020). Interestingly, GDRO-TINF achieves a $10^{-4}$ optimality gap in $T = 10^6$ iterations, which is faster than the theoretical $T^{-0.5}$ rate in Theorem 3.

We perform additional experiments in the deep learning regime across five benchmark datasets from WILDS (Sagawa et al., 2020) including Waterbirds, FMOW, MultiNLI, etc. Worst group and average test performance of various methods is reported in Appendix E.2

## 6 CONCLUSION

In this work we settle the optimal achievable regret in the group DRO problem, up to a log factor, by (1) developing a new technique that enables us to employ online optimization techniques in offline robust optimization, and (2) combining the right ingredients from online adversarial algorithms to achieve the almost best rate for group DRO. We hope that our work further encourages researchers in the future to employ such reductions from online to offline optimization. Besides the demonstrated theoretical guarantees, our extensive experiments on real and synthetic data illustrate that our algorithm is competitive with state-of-the-art methods.

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

# A PRELIMINARIES OF ONLINE CONVEX OPTIMIZATION AND NO-REGRET DYNAMICS

In this section, we briefly introduce necessary results from online convex optimization (OCO). For the further details of OCO, refer to Hazan (2016); Orabona (2019).

## A.1 REGRET BOUNDS OF OCO ALGORITHMS

Let $X \subseteq \mathbb{R}^d$ be a compact convex set and $\Psi : X \to \mathbb{R}$ be a strictly convex function such that $\|\partial \Psi(x)\| \to +\infty$ as $x \to \partial X$. Online mirror descent (OMD) is the following online learning algorithm. For $t = 1, \ldots, T$:

1. Let $\tilde{x}_{t+1} \in \mathbb{R}^n$ be the solution of $\nabla \Psi(\tilde{x}_{t+1}) = \nabla \Psi(x_t) - \eta_t \nabla_t$, where $\eta_t > 0$ is a step size and $\nabla_t = \nabla f_t(x_t)$ is the gradient feedback of round $t$.

2. Let $x_{t+1} \in \arg \min_{x \in X} D_\Psi(x, \tilde{x}_{t+1})$, where $D_\Psi(x, y) = \Psi(x) - \Psi(y) - \nabla \Psi(y)^\top (x - y)$ is the Bregman divergence with respect to $\Psi$.

We use the following regret bound.

**Lemma 1** (Regret Bound of OMD; see, e.g., Orabona (2019, Theorem 6.8)). *OMD satisfies that for any $x^* \in X$,*

$$\sum_{t=1}^{T} f_t(x_t) - \sum_{t=1}^{T} f_t(x^*) \leq \frac{1}{2} \sum_{t=1}^{T} \eta_t \|\nabla_t\|_{t,*}^2 + \frac{D_\Psi(x^*, x_1)}{\eta_1} + \sum_{t=2}^{T} \left( \frac{1}{\eta_t} - \frac{1}{\eta_{t-1}} \right) D(x^*, x_t), \quad (6)$$

*where $\|x\|_t$ denotes the local norm, i.e., $\|x\|_t := \sqrt{x^\top \nabla^2 \Psi(z_t) x}$ for some $z_t \in [x_t, \tilde{x}_{t+1}]$ and $\|x\|_{t,*} := \sqrt{x^\top \nabla^2 \Psi(z_t)^{-1} x}$ is its dual norm.*

In this paper, we use regret bounds for the following specific choices of $\Psi$.

**Online Gradient Descent** OMD for $\Psi(x) = \frac{1}{2} \|x\|_2^2$ on a generic compact convex set $X$ is simply online gradient descent (OGD) Zinkevich (2003):

$$x_{t+1} = \text{proj}_X (x_t - \eta_t \nabla_t).$$

Note that $D(x, y) = \frac{1}{2} \|x - y\|_2^2$ and the minimizing the Bregman divergence is given by orthogonal projection.

**Lemma 2** (Regret Bound of OGD). *OMD satisfies that for any $x^* \in X$,*

$$\sum_{t=1}^{T} f_t(x_t) - \sum_{t=1}^{T} f_t(x^*) \leq \frac{1}{2} \sum_{t=1}^{T} \eta_t \|\nabla_t\|^2 + \frac{\|x^* - x_1\|_2^2}{2\eta_1} + \frac{1}{2} \sum_{t=2}^{T} \left( \frac{1}{\eta_t} - \frac{1}{\eta_{t-1}} \right) \|x^* - x_t\|_2^2. \quad (7)$$

*If we use decreasing step sizes and $\max_{t=1}^{T} \|x^* - x_t\| \leq D$, we have*

$$\sum_{t=1}^{T} f_t(x_t) - \sum_{t=1}^{T} f_t(x^*) \leq \frac{1}{2} \sum_{t=1}^{T} \eta_t \|\nabla_t\|^2 + \frac{D^2}{2\eta_T}. \quad (8)$$

**Hedge** OMD for $\Psi(x) = \sum_i (x_i \log x_i - x_i)$ on the probability simplex is the Hedge algorithm.

$$\tilde{x}_{t+1} = x_t \exp(-\eta_t \nabla_t), \quad x_{t+1} = \frac{\tilde{x}_{t+1}}{\|\tilde{x}_{t+1}\|_1}.$$

Note that $\nabla^2 \Psi(x) = \text{diag}(1/x_i)$. If $\nabla_t \geq 0$, then $\tilde{x}_{t+1} \leq x_t$ and $\|\nabla_t\|_{t,*} \leq \|\nabla_t\|_{\nabla^2 \Psi(x_t)^{-1}}$. For $x_1 = \mathbf{1}/d$, $D(x^*, x_1) \leq \log d$ for any $x^*$.

**Lemma 3** (Regret Bound of Hedge). *For $\nabla_t \geq 0$ ($t = 1, \ldots, 1$), Hedge with fixed step size $\eta > 0$ satisfies*

$$\sum_{t=1}^{T} f_t(x_t) - \sum_{t=1}^{T} f_t(x^*) \leq \frac{\eta}{2} \sum_{t=1}^{T} \|\nabla_t\|_{\nabla^2 \Psi(x_t)^{-1}}^2 + \frac{\log d}{\eta}. \quad (9)$$

**Tsallis-INF**   OMD for $\Psi(x) = 2(1 - \sum_i \sqrt{x_i})$ on the probability simplex is the Tsallis-INF algorithm:

$$\tilde{x}_{t+1} \leftarrow x_t \left(\mathbf{1} - \eta_t \nabla_t\right)^{-2}, \quad x_{t+1} = \left(\frac{1}{\sqrt{\tilde{x}_{t+1}}} - \alpha\mathbf{1}\right)^{-2},$$

where $\alpha$ is the scaling factor such that $x_{t+1}$ is in the probability simplex. Note that if $\nabla_t \geq 0$, then $\tilde{x}_{t+1} \leq x_t$ and $\|\nabla_t\|_{t,*} \leq \|\nabla_t\|_{\nabla^2\Psi(x_t)^{-1}}$ as in Hedge. For $x_1 = \mathbf{1}/d$, $D(x^*, x_1) \leq \sqrt{d}$ for any $x^*$.

**Lemma 4** (Regret Bound of Tsallis-INF). *For $\nabla_t \geq 0$ ($t = 1, \ldots, 1$), Tsallis-INF with fixed step size $\eta > 0$ satisfies*

$$\sum_{t=1}^{T} f_t(x_t) - \sum_{t=1}^{T} f_t(x^*) \leq \frac{\eta}{2} \sum_{t=1}^{T} \|\nabla_t\|^2_{\nabla^2\Psi(x_t)^{-1}} + \frac{\sqrt{d}}{\eta}. \tag{10}$$

### A.2   CONVERGENCE OF NO-REGRET DYNAMICS

Let us write DRO equation 1 as

$$\min_{\theta \in \Theta} \max_{q \in Q} L(\theta, q).$$

Note that $L(\theta, q)$ is convex in $\theta$ and linear in $q$.

Let us assume that we apply stochastic no-regret dynamics to this minimax problem. The $\theta$-player and $q$-player run online algorithms on $\Theta$ and $Q$, respectively. The feedback to $\theta$-player and $q$-player are $\hat{\nabla}_{\theta,t}$ and $\hat{\nabla}_{q,t}$, respectively, which are unbiased gradient estimators of $L$. We can analyze the optimality gap of stochastic no-regret dynamics using the regrets. Let $\theta^*$ be an optimal solution and

$$\epsilon_T := \max_{q \in Q} L(\bar{\theta}_{1:T}, q) - \max_{q \in Q} L(\theta^*, q).$$

be the optimality gap of the averaged iterate $\bar{\theta}_{1:T} = \frac{1}{T}\sum_{t=1}^{T} \theta_t$. Let

$$R_\theta(T; \theta^*) = \sum_{t=1}^{T} L(\theta_t, q_t) - \sum_{t=1}^{T} L(\theta^*, q_t)$$

$$R_q(T) = \max_{q \in \Delta_m} \sum_{t=1}^{T} L(\theta_t, q) - \sum_{t=1}^{T} L(\theta_t, q_t)$$

be regrets of the $\theta$-player and $q$-player, respectively. Then, by the definition of regret and Jensen's inequality, we have

$$\epsilon_T \leq \max_{q \in Q} \frac{1}{T} \sum_{t=1}^{T} L(\theta_t, q) - \max_{q \in Q} L(\theta^*, q)$$

$$= \frac{R_q(T)}{T} + \frac{1}{T} \sum_{t=1}^{T} L(\theta_t, q_t) - \max_{q \in Q} L(\theta^*, q)$$

$$\leq \frac{R_q(T)}{T} + \frac{1}{T} \sum_{t=1}^{T} L(\theta_t, q_t) - \frac{1}{T} \sum_{t=1}^{T} L(\theta^*, q_t)$$

$$= \frac{R_q(T) + R_\theta(T; \theta^*)}{T}.$$

Therefore,

$$\mathbb{E}[\epsilon_T] \leq \frac{\mathbb{E}[R_q(T) + R_\theta(T; \theta^*)]}{T}, \tag{11}$$

where the expectation is taken over the randomness of gradient estimators and the algorithm.

Since $\epsilon_T(q^*) \leq \epsilon_T$ for any fixed $q^* \in Q$, we also have

$$\mathbb{E}[\epsilon_T(q^*)] \leq \frac{\mathbb{E}[R_q(T; q^*) + R_\theta(T; \theta^*)]}{T} \tag{12}$$

for any fixed $q^*$, where

$$R_q(T; q^*) = \sum_{t=1}^{T} L(\theta_t, q^*) - \sum_{t=1}^{T} L(\theta_t, q_t)$$

is the regret of the $q$-player with respect to fixed $q^* \in Q$.

## A.3 EXP3P

To analyze the convergence rate using the above bound, we need to bound the expected regret of the $q$-player for an *adaptive* adversary. This is not possible by OMD bounds because it only considers fixed optimal solutions, i.e., an *oblivious* adversary. Thankfully, in group DRO, we can use the EXP3P algorithm Auer et al. (2003), which has desired regret bounds for adaptive adversaries.

---

**Algorithm 4** EXP3P

---

**Require:** parameters $\beta, \eta, \gamma > 0$
1: Let $q_1 = (1/m, \ldots, 1/m)$.
2: **for** $t = 1, \ldots, T$ **do**
3:     Sample $i_t \sim q_t$.
4:     Let $\tilde{g}_t := \frac{\ell_{i_t}\mathbf{e}_{i_t} + \beta\mathbf{1}}{q_{i_t}}$, $G_t := \sum_{\tau=1}^{t} \tilde{g}_t$, and $Z = \sum_{i\in[m]} \exp(\eta G_{t,i})$.
5:     $q_{t+1} \leftarrow (1-\gamma)\frac{\exp(\eta G_t)}{Z} + \frac{\gamma\mathbf{1}}{m}$.

---

**Theorem 6** (see, e.g., (Bubeck et al., 2012, Theorem 3.4)). *Let* $g_t \in [0,1]^m$ *for* $t \in T$. *For* $\beta = \sqrt{\frac{\log m}{mT}}$, $\eta = O(\sqrt{\frac{\log m}{mT}})$, *and* $\gamma = O(\sqrt{\frac{m\log m}{T}})$, *EXP3P achieves*

$$\mathbb{E}[R_q(T)] = \mathbb{E}\left[\max_{i^*\in[m]} \sum_{t=1}^{T}(g_{t,i^*} - g_{t,i_t})\right] \lesssim \sqrt{mT\log m}.$$

# B OMMITED PROOFS

## B.1 PROOF OF THEOREM 1

Let $I_t$ and $z_t$ be the chosen group and the sample at iteration $t$, respectively. Observe that Algorithm 1 is stochastic no-regret dynamics with OGD, OMD, and gradient estimators

$$\hat{\nabla}_{\theta,t} := \nabla_\theta \ell(\theta_t; z_t), \quad \hat{\nabla}_{q,t} := \frac{1}{q_{t,I_t}}\ell(\theta_t; z_t)\mathbf{e}_{I_t}.$$

For OGD, we use Lemma 2. We have $\|\hat{\nabla}_{\theta,t}\|_2^2 \leq G$ by assumption. Therefore,

$$\mathbb{E}[R_\theta(T)] \leq \frac{G^2}{2}\sum_{t=1}^{T}\eta_{\theta,t} + \frac{D^2}{2\eta_{\theta,T}}$$

by Lemma 2. For OMD, we use Lemma 1. Since $\hat{\nabla}_{q,t} = \frac{1}{q_{t,I_t}}\ell(\theta_t, z_t)\mathbf{e}_{I_t}$, we obtain

$$\|\hat{\nabla}_{q,t}\|_{\nabla^2\Psi(q_t)^{-1}}^2 = \frac{\ell(\theta_t, z_t)^2(\nabla^2\Psi(q_t)^{-1})_{I_t,I_t}}{q_{t,I_t}^2}$$
$$\leq \frac{M^2(\nabla^2\Psi(q_t)^{-1})_{I_t,I_t}}{q_{t,I_t}^2}.$$

Hence we obtain from Lemma 1,

$$\mathbb{E}[R_q(T; q^*)] \leq \frac{1}{2}\sum_{t=1}^{T}\eta_q\mathbb{E}_{I_t}\left[\frac{(\nabla^2\Psi(q_t))_{I_t,I_t}^{-1}}{q_{t,I_t}^2}\right] + \frac{D_\Psi(q^*, q_1)}{\eta_q}$$

for any $q^* \in Q$. Now the theorem is immediate from equation 12.

### B.2    PROOF OF THEOREM 2

Observe that Algorithm 2 is stochastic no-regret dynamics with OGD, EXP3P, and the same gradient estimators as above. Without loss of generality, we can assume $M = 1$; general case follows scaling the loss functions accordingly. Using the regret bound in Theorem 6, we have

$$\mathbb{E}[\epsilon_q(T)] \lesssim \sqrt{mT \log m}.$$

Now the theorem follows from equation 11.

### B.3    PROOF OF THEOREM 3

Observe that Algorithm 3 is stochastic no-regret dynamics with OGD, Tsallis-INF, and the same gradient estimators as above. From Theorem 1, it suffices to bound the local norm with respect to the Tsallis entropy regularizer. Observe that $\nabla^2 \Psi(q_t) = \frac{1}{2} \operatorname{diag}(q_t^{-2/3})$. Conditioned on $I_1, \ldots, I_{t-1}$, we have

$$
\begin{aligned}
\mathbb{E}_{I_t} \left[ \frac{(\nabla^2 \Psi(q_t))^{-1}_{I_t, I_t}}{q_{t, I_t}^2} \right] &= \sum_{i=1}^m \Pr(I_t = i) \cdot \frac{2}{q_{t,i}^{1/2}} \\
&\leq 2 \sum_{i=1}^m q_{t,i}^{1/2} \\
&\leq 2\sqrt{m} \sqrt{\sum_{i=1}^m q_{t,i}} \\
&= 2\sqrt{m}.
\end{aligned}
$$

By Theorem 1, we obtain equation 5.

## C    OMMITED PROOFS IN SECTION 4

In this section, we prove Theorem 5.

We show that the minimax optimality gap is $\Omega(GD/\sqrt{T})$ and $\Omega(M\sqrt{m/T})$, separately. The first lower bound is immediate from the well-known lower bound of stochastic convex optimization (see, e.g., Agarwal et al. (2012)). Hence, it suffices to show the second lower bound.

Note that it suffices to show the lower bound for a constant $M$; below we construct instances with $M = 2$. The general case follows by scaling the objective with $M$. Consider the following instance of group DRO which we construct with respect to an $m$-dimensional vector $\mu = (\mu_1, \ldots, \mu_m) \in [0, 1]^m$ of Bernoulli biases. Let $\Theta$ be the unit interval $[0, 1]$. Let

$$\ell(\theta; Z) = Z_1 f_1(\theta) + Z_2 f_2(\theta) + Z_3,$$

where $f_1(\theta) = \delta\theta$ and $f_2(\theta) = \delta(1 - \theta)$ are linear functions over the interval $[0, 1]$ and $\delta > 0$ is the accuracy parameter determined later. We define a joint distribution $P_i$ of $Z$ as follows: for $i = 1, \ldots, m - 1$, let

$$
P_i : \begin{cases} Z_1 = 0 \text{ a.s.} \\ Z_2 = 1 \text{ a.s.} \\ Z_3 \sim \operatorname{Ber}(\mu_i) \end{cases}
$$

where $a.s.$ stands for almost surely. For the last group distribution $i = m$, let

$$
P_m : \begin{cases} Z_1 = 1 \text{ a.s.} \\ Z_2 = 0 \text{ a.s.} \\ Z_3 \sim \operatorname{Ber}(\mu_m). \end{cases}
$$

Then,

$$
\mathbb{E}_{Z \sim P_i}[\ell(\theta; Z)] = \begin{cases} \delta(1 - \theta) + \mu_i & (i = 1, \ldots, m - 1) \\ \delta\theta + \mu_m. & (i = m) \end{cases}
$$

The information of an outcome of a single stochastic oracle call to $P_i$ is no more than that of a single sample of the $i$th Bernoulli distribution $\mathrm{Ber}(\mu_i)$.

Let us fix $\hat{\theta} \in \mathcal{A}_T$ arbitrarily. Let $\mathcal{P}_0$ be the set of distributions $(P_i)$ constructed as above with

$$\mu^0 = (1/2, 1/2, \ldots, 1/2).$$

It is clear that $\min_{\theta^* \in \Theta} \max_{P \in \mathcal{P}_0} \mathbb{E}_{Z \sim P}[\ell(\theta^*; Z)] = 1/2 + \delta/2$, which is attained by $\theta^* = 1/2$. We denote by $Q_0$ the distribution of the outcomes of stochastic oracles observed by $\hat{\theta}$ under $\mathcal{P}_0$. Furthermore, let $T_i$ be the expected number of queries to the $i$th stochastic oracle made by $\hat{\theta}$ under $\mathcal{P}_0$. Since $\hat{\theta}$ makes $T$ queries in total, there exists $i^* \neq m$ such that $T_{i^*} \leq \frac{T}{m-1}$. Let $\mathcal{P}_1$ be the set of distributions constructed as above with

$$\mu^1 = (1/2, 1/2, \ldots, 1/2, \overset{i^*}{1/2} + \delta, 1/2, \ldots, 1/2).$$

**Lemma 5.**
$$\max\{R(\theta, \ell, \mathcal{P}_0), R(\theta, \ell, \mathcal{P}_1)\} \geq \delta/4$$

*for any $\theta$.*

*Proof.* We consider two different cases: $\theta \geq \frac{3}{4}$ and $\theta < \frac{3}{4}$.

For $\theta \geq 3/4$, we have $R(\theta, \ell, \mathcal{P}_0) \geq \frac{\delta}{4}$ since

$$\max_{P \in \mathcal{P}_0} \mathbb{E}_{Z \sim P}[\ell(\theta^*; Z)] = \max_{P \in \mathcal{P}_0} \mathbb{E}_{Z \sim P}[\ell(1/2; Z)] = \delta/2 + 1/2,$$

while

$$\max_{P \in \mathcal{P}_0} \mathbb{E}_{Z \sim P}[\ell(\theta; Z)] = \mathbb{E}_{Z \sim P_m}[\ell(\theta; Z)] \geq \delta/2 + \delta/4 + 1/2.$$

For the other case, $\theta < 3/4$, we show that $R(\theta, \ell, \mathcal{P}_1) \geq \frac{\delta}{4}$. This holds as

$$\max_{P \in \mathcal{P}_1} \mathbb{E}_{Z \sim P}[\ell(\theta^*; Z)] = \max_{P \in \mathcal{P}_1} \mathbb{E}_{Z \sim P}[\ell(1; Z)] = \delta + 1/2,$$

while

$$\max_{P \in \mathcal{P}_1} \mathbb{E}_{Z \sim P}[\ell(\theta; Z)] = \mathbb{E}_{Z \sim P_{i^*}}[\ell(\theta; Z)] \geq \delta + \delta/4 + 1/2.$$

This completes the proof. $\qquad\square$

We denote by $Q_1$ the distribution of the outcomes of stochastic oracles observed by $\hat{\theta}$ under $\mathcal{P}_1$. By Lecam's two-point method,

$$\inf_{\hat{\theta}} \sup_{\mathcal{P}} \mathbb{E}[R(\hat{\theta}, \ell, \mathcal{P})] \geq \frac{\delta}{2}(1 - d_{\mathrm{TV}}(Q_0, Q_1)),$$

where the expectation is taken over the outcomes of the stochastic oracle and $d_{\mathrm{TV}}$ denotes the total variation distance. We proceed to bound the right-hand side. By the Pinsker inequality,

$$d_{\mathrm{TV}}(Q_0, Q_1)^2 \lesssim D_{\mathrm{KL}}(Q_0 \,\|\, Q_1),$$

where $D_{\mathrm{KL}}$ denotes the Kullback-Leibler divergence. By the standard computation, we can show the following.

**Lemma 6.**
$$D_{\mathrm{KL}}(Q_0 \,\|\, Q_1) \lesssim \delta^2 T_{i^*} \leq \frac{\delta^2 T}{m-1}.$$

*for $\delta \in (0, 1/4)$,*

Thus, setting $\delta = O(\sqrt{m/T})$, we obtain

$$\inf_{\hat{\theta} \in \mathcal{A}_T} \sup_{\ell \in \mathcal{L}, \mathcal{P}} R(\hat{\theta}, \ell, \mathcal{P}) \gtrsim \sqrt{\frac{m}{T}},$$

which completes the proof of Theorem 5.

## C.1 PROOF OF LEMMA 6

Now we prove Lemma 6 for the completeness. Let $o_t$ be the outcome of the $t$th query to the stochastic oracle. We will use the shorthand notation $o_{1:t}$ to denote the outcomes $(o_1, \ldots, o_t)$ up to the $t$th queries. Let $I_t \in [m]$ be the index of stochastic oracles that $\hat{\theta}$ queries in the $t$th round. Note that $I_t$ is determined by $o_{1:t-1}$. Then, we have

$$
\begin{aligned}
D_{\mathrm{KL}}(Q_0 \parallel Q_1) &= \sum_{t=1}^{T} D_{\mathrm{KL}}(Q_0(o_t \mid o_{1:t-1}) \parallel Q_1(o_t \mid o_{1:t-1})) && \text{(chain rule)} \\
&\leq \sum_{t=1}^{T} \mathbb{E}_{o_{1:t-1} \sim Q_0} \left[ D_{\mathrm{KL}}(\mathrm{Ber}(\mu_{I_t}^0) \parallel \mathrm{Ber}(\mu_{I_t}^1)) \right] && \text{(data-processing inequality)} \\
&= \sum_{t=1}^{T} \mathbb{E}_{o_{1:t-1} \sim Q_0} \left[ \mathbf{1}[I_t = i^*] D_{\mathrm{KL}}(\mathrm{Ber}(1/2) \parallel \mathrm{Ber}(1/2 + \delta)) \right] \\
&= T_{i^*} \cdot D_{\mathrm{KL}}(\mathrm{Ber}(1/2) \parallel \mathrm{Ber}(1/2 + \delta)).
\end{aligned}
$$

Furthermore, for $\delta \in (0, 1/4)$,

$$
\begin{aligned}
D_{\mathrm{KL}}(\mathrm{Ber}(1/2) \parallel \mathrm{Ber}(1/2 + \delta)) &= \frac{1}{2} \log \frac{1/2}{1/2 + \delta} + \frac{1}{2} \log \frac{1/2}{1/2 - \delta} \\
&= \frac{1}{2} \log \left( 1 - \frac{2\delta}{1 + 2\delta} \right) + \frac{1}{2} \log \left( 1 + \frac{2\delta}{1 - 2\delta} \right) \\
&\leq -\frac{\delta}{1 + 2\delta} + \frac{\delta}{1 - 2\delta} \\
&\leq \frac{4\delta^2}{(1 + 2\delta)(1 - 2\delta)} \leq 8\delta^2.
\end{aligned}
$$

This completes the proof.

## D ALGORITHM OF SAGAWA ET AL. FOR GROUP DRO

Here we present the algorithm by Sagawa et al. (2020) for group DRO. Algorithm 5 shows the pseudocode. In each iteration $t$, the algorithm picks group index $i_t \in [m]$ uniformly at random and obtains an i.i.d. sample $z \sim P_{i_t}$. Then, the algorithm performs one step of projected gradient descent and Hedge on $\theta_t \in \Theta$ and $q_t \in \Delta_m$, respectively, where the gradients are estimated with $i_t$ and $z$. Note that $q_t$ is only used for the scaling factor of the gradient estimator. In each iteration, the algorithm performs a single orthogonal projection onto $\Theta$ and $O(m + n)$ operations to update $\theta_t, q_t$.

---

**Algorithm 5** Algorithm of Sagawa et al.

---

**Require:** initial solution $\theta_1 \in \Theta$, number of iteration $T$, and step sizes $\eta_{\theta,t} > 0$ $(t \in [T])$, $\eta_q > 0$.
1: Let $q_t = (1/m, \ldots, 1/m)$.
2: **for** $t = 1, \ldots, T$ **do**
3:      Sample $i_t \sim [m]$ uniformly at random.
4:      Call the stochastic oracle to obtain $z \sim P_{i_t}$.
5:      $\theta_{t+1} \leftarrow \mathrm{proj}_\Theta(\theta_t - m q_{t,i_t} \eta_{\theta,t} \nabla_\theta \ell(\theta_t; z))$
6:      $\tilde{q}_{t+1} \leftarrow q_t \exp(m \eta_q \ell(\theta_t; z) \mathbf{e}_{i_t})$ and $q_{t+1} \leftarrow \frac{\tilde{q}_{t+1}}{\sum_i \tilde{q}_{t+1,i}}$.
7: **return** $\frac{1}{T} \sum_{t=1}^{T} \theta_t$.

---

In the view of no-regret dynamics, the main difference between our algorithms and Sagawa et al. (2020) is the gradient estimators; see Table 2.

Table 2: Algorithms as stochastic no-regret dynamics

| | $\mathcal{A}_\theta$ | $\mathcal{A}_q$ | $\hat{\nabla}_{\theta,t}, \hat{\nabla}_{q,t}$ |
|---|---|---|---|
| Algorithm 5 | OGD | Hedge | $\hat{\nabla}_{\theta,t} := m q_{t,i} \nabla_\theta \ell(\theta; z), \quad \hat{\nabla}_{q,t} := m\ell(\theta_t; z)\mathbf{e}_i. \quad (i \sim [m], z \sim P_i)$ |
| Algorithm 2 | OGD | EXP3P | $\hat{\nabla}_{\theta,t} := \nabla_\theta \ell(\theta_t; z), \quad \hat{\nabla}_{q,t} := \dfrac{1}{q_{t,i}}\ell(\theta_t; z)\mathbf{e}_i. \quad (i \sim q_t, z \sim P_i)$ |
| Algorithm 3 | OGD | Tsallis-INF | $\hat{\nabla}_{\theta,t} := \nabla_\theta \ell(\theta_t; z), \quad \hat{\nabla}_{q,t} := \dfrac{1}{q_{t,i}}\ell(\theta_t; z)\mathbf{e}_i. \quad (i \sim q_t, z \sim P_i)$ |

# E ADDITIONAL EXPERIMENTS

## E.1 EXPERIMENT WITH SYNTHETIC DATASET FOR CONVEX REGIME

**Dataset.** To observe the performance of the algorithms over the regime of high-dimension model parameters and the larger number of groups, we also conducted experiments using the following synthetic instances. First, we set $n = 500$ and varied $m \in \{10, 50, 100\}$. For each group $i \in [m]$, we generated the true classifier $\theta_i^* \in \mathbb{R}^n$ from the uniform distribution over the unit sphere in $\mathbb{R}^n$. The $i$th group distribution $P_i$ was the empirical distribution of 1,000 data points, where each data point $(a, b)$ was drawn as $a \sim N(0, I_n)$ and $b = \text{sign}(a^\top \theta_i^*)$ with probability 0.9 and $b = -\text{sign}(a^\top \theta_i^*)$ with probability 0.1. We trained the linear model with the hinge loss function. Finally, the group-DRO objective is

$$\max_{i=1}^{m} \mathbb{E}_{(a,b)\sim P_i}[\ell(\theta; a, b)].$$

The feasible region is the Euclidean ball of radius $D = 10$.

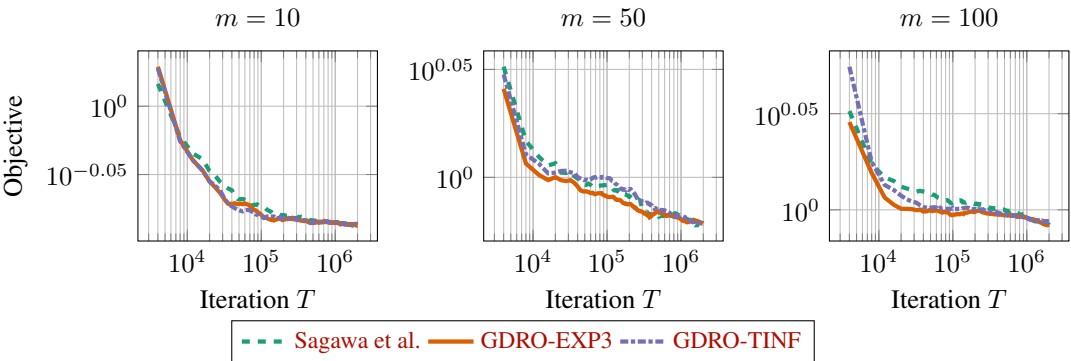

Figure 2: Results on the synthetic dataset for the convex regime. Both axes are log-scale

**Result.** In Figure 2, we plot the objective values of the averaged iterate against the number of iterations. For all the values of $m$, our algorithms (especially GDRO-EXP3) consistently achieve smaller loss values faster than the known algorithm. The performance gap between our algorithms and the known algorithm increased as $m$ grows, which verifies that our algorithms have better dependence on $m$ in the convergence rate.

## E.2 EXPERIMENTS IN THE DEEP LEARNING REGIME

Our convergence analysis focuses on the convex regime. However, algorithms designed for the convex regime often work well even for the deep learning regime. Here, we compare our algorithms with the known algorithms in the deep learning regime.

**Dataset.** We used Wilds (Koh et al., 2021), which consists of various real-world data for machine learning tasks and various baseline optimization algorithms. Each task specifies the loss function, performance metric, train-test data split, and neural net architecture. We used Waterbirds, CIVIL-Comments, FMoW-Wilds, PovertyMAP-Wilds from Wilds. For example, Waterbirds consists of

Table 3: Worst-group test performance of algorithms for Wilds datasets. S. et al., EXP, and TINF denote the algorithm of Sagawa et al. (2020), GDRO-EXP, and GDRO-TINF, respectively. The value format is mean $\pm$ standard deviation. The best mean in each row is in bold.

| Dataset | Metric | Heuristics | ERM | S. et al. | EXP | TINF |
|---|---|---|---|---|---|---|
| Waterbirds | accuracy | standard training | $58.9 \pm 3.21$ | $75.5 \pm 1.66$ | $70.6 \pm 4.07$ | $\mathbf{77.4 \pm 2.31}$ |
| | | penalty | $24.3 \pm 2.43$ | $\mathbf{84.9 \pm 1.19}$ | $77.4 \pm 0.72$ | $79.0 \pm 0.71$ |
| | | early stop + penalty | $9.2 \pm 1.34$ | $86.2 \pm 0.96$ | $85.4 \pm 1.44$ | $\mathbf{87.5 \pm 2.68}$ |
| MultiNLI | accuracy | standard training | $65.1 \pm 1.22$ | $66.4 \pm 1.84$ | $\mathbf{72.4 \pm 0.83}$ | $72.2 \pm 2.47$ |
| | | early stop + penalty | $66.8 \pm 1.56$ | $77.7 \pm 3.69$ | $76.9 \pm 2.25$ | $\mathbf{78.1 \pm 1.63}$ |
| CIVIL-Comments | accuracy | standard training | $58.4 \pm 1.29$ | $67.5 \pm 3.43$ | $62.5 \pm 1.47$ | $\mathbf{68.8 \pm 1.34}$ |
| FMOW-Wilds | accuracy | standard training | $\mathbf{33.4 \pm 0.53}$ | $31.9 \pm 1.22$ | $31.7 \pm 1.21$ | $33.31 \pm 0.83$ |
| PovertyMAP-Wilds | U/R Pearson R | standard training | $0.413 \pm 0.025$ | $0.422 \pm 0.031$ | $\mathbf{0.491 \pm 0.037}$ | $0.439 \pm 0.046$ |

Table 4: Average test performance of algorithms for Wilds datasets. S. et al., EXP, and TINF denote the algorithm of Sagawa et al. (2020), GDRO-EXP, and GDRO-TINF, respectively. The value format is mean $\pm$ standard deviation. The best mean in each row is in bold.

| Dataset | Metric | Heuristics | ERM | S. et al. | EXP | TINF |
|---|---|---|---|---|---|---|
| Waterbirds | accuracy | standard training | $\mathbf{97.2 \pm 0.35}$ | $90.5 \pm 0.21$ | $91.3 \pm 0.23$ | $91.1 \pm 0.52$ |
| | | penalty | $\mathbf{94.5 \pm 0.78}$ | $92.2 \pm 0.42$ | $90.0 \pm 0.40$ | $90.9 \pm 1.50$ |
| | | early stop + penalty | $\mathbf{92.4 \pm 0.18}$ | $90.0 \pm 0.70$ | $91.4 \pm 0.74$ | $90.6 \pm 1.42$ |
| MultiNLI | accuracy | standard training | $\mathbf{81.8 \pm 0.45}$ | $82.0 \pm 2.78$ | $80.4 \pm 1.20$ | $81.4 \pm 0.79$ |
| | | early stop + penalty | $\mathbf{82.4 \pm 0.93}$ | $81.4 \pm 2.30$ | $81.3 \pm 0.84$ | $82.0 \pm 0.85$ |
| CIVIL-Comments | accuracy | standard training | $\mathbf{92.7 \pm 0.61}$ | $90.2 \pm 0.49$ | $91.2 \pm 0.58$ | $90.2 \pm 0.90$ |
| FMOW-Wilds | accuracy | standard training | $\mathbf{52.1 \pm 0.27}$ | $51.7 \pm 0.39$ | $51.2 \pm 0.39$ | $51.9 \pm 0.46$ |
| PovertyMAP-Wilds | U/R Pearson R | standard training | $0.725 \pm 0.019$ | $0.711 \pm 0.047$ | $\mathbf{0.814 \pm 0.021}$ | $0.755 \pm 0.031$ |

images of two kinds of birds (landbirds and waterbirds) with different backgrounds (land and water) and the task is to predict the types of birds in images. For further detail, see Appendix E.3 and their original paper (Koh et al., 2021).

**Algorithm.** We implemented GDRO-EXP3P and GDRO-TINF within the Python framework of Wilds. As baseline methods, we used empirical risk minimization (ERM) and the algorithm of Sagawa et al. (2020) provided by Wilds. We used the standard neural network architecture specified by Wilds for our learning models; for example, ResNet50 for Waterbirds and BERT for MultiNLI and CIVIL-Comments, etc. For $\theta$-player algorithms, we used the default optimizer with default hyperparameters in Wilds for all algorithms. We used the official data split provided by Wilds. We trained each model with the default number of epochs (e.g., 200 epochs for Waterbirds) in Wilds and report the performance of the best iterate.

**Optimization heuristics.** Sagawa et al. (2020) proposed several optimization heuristics, which were shown to improve the performance in their Waterbirds experiment. To complement our experiments, we also report the results using these optimization heuristics in the Waterbirds and MultiNLI experiments. In particular, we run Vanilla SGD (standard), $\ell_2$-regularization (penalty), and both early stopping and $\ell_2$-regularization (early stop+penalty).

**Step sizes.** For ERM and the algorithm Sagawa et al. (2020), we used the default setting provided by Wilds. Our algorithms used the following settings. For $\theta$-player algorithms, we used the default optimizer with default hyperparameters in Wilds. For $q$-player algorithms (EXP3P and TINF), we used the default step size $\eta_q = 0.01$ for the algorithm of Sagawa et al. (2020) in Wilds.

**Mini-batch.** We found that the following mini-batch strategy yielded the best performance. Each mini-batch consists of $B$ samples constructed as follows: A batch of $B$ elements is sampled from the training dataset according to the sampling strategy of the $q$-player. Corresponding to the indices of the sampled groups in the batch, data points are selected at random. After constructing the mini-batch, we then update the model parameter $\theta_t$ and group weight $q_t$ using the gradient and loss

averaged over the mini-batch for each group separately. We set $B$ to the default mini-batch size provided by Wilds (e.g., $B = 128$ for Waterbirds) in our experiments.

### E.2.1 RESULTS

We report the worst group and average test performance for each dataset in Wilds in Tables 3 and 4 respectively. Here, the mean and standard deviation (stddev) are computed from three independent runs with different random seeds. In almost all datasets, GDRO-EXP3P and GDRO-TINF consistently achieved the best worst-group accuracy. Although the performances of the algorithms except ERM are relatively close, remark that we did not tune the step size for GDRO-EXP and GDRO-TINF but used the default step size in Wilds, which is tuned for the algorithm of Sagawa et al. (2020).

### E.3 DETAILS OF EXPERIMENTS IN DEEP LEARNING REGIME

We summarize the characteristics of Wilds tasks we used in our experiments in the deep learning regime. The full details can be found in Koh et al. (2021).

**Waterbirds.** The Waterbirds dataset consists of images of birds of two kinds (waterbirds and landbirds) with different backgrounds (land and water). The task is to predict the type of birds in images. There are $m = 4$ groups corresponding to the combinations of birds and backgrounds. The number of training examples is 4795 in total and 56 in the smallest group (waterbirds on land). We used ResNet50 as our learning model. We used the torch-vision implementation of ResNet50 as suggested in Wilds.

**MultiNLI.** The MultiNLI dataset is a natural language dataset consisting of labeled sentences. We used the modified version of MultiNLI provided by Sagawa et al. (2020)[2]. Each image is assigned to $m = 6$ groups corresponding to the combination of labels {entailed, neutral, contradictory} and the existence of negation words {no negation, negation}. The training set contains 206175 examples with 1521 examples in the smallest group (entailment with negations). We used Hugging Face pytorch-transformers implementation of the BERT with pre-trained weights.

**CIVIL-Comments.** CIVIL-Comments is a natural language dataset of distribution shifts with different demographic identities. The task is to predict whether a given text is toxic or not. There are $m = 2$ groups (toxic or not). The learning model is BERT same as MultiNLI.

**FMoW-wilds.** FMoW-wilds consists of RGB satellite images of 224×224 pixels. Each image has its label (use or land) and geographical region (Africa, the Americas, Oceania, Asia, or Europe). The task is to predict the label of a given image. There are $m = 8$ groups (the year where each image was taken). The learning model is DenseNet121.

**PovertyMAP-wilds.** PovertyMAP-Wilds consists of LandSat satellite image with 8 channels (resized to 224 x 224 pixels) with a label of real-valued asset wealth index. The task is to predict the label of a given image. There are $m = 8$ groups (the country where each image was taken). The learning model is Resnet18ms.

---

[2]https://github.com/kohpangwei/group_DRO

