# OpenReview forum: "Optimal algorithms for group distributionally robust optimization and beyond"
_ICLR.cc/2024/Conference — ICLR 2024 Conference Withdrawn Submission_

### Official Review · Reviewer_xpZv · 2023-10-28

**Soundness:** 2 fair
**Presentation:** 1 poor
**Contribution:** 3 good
**Rating:** 5
**Confidence:** 4

**Summary:**

This paper studies the generalized Group DRO and provides matching upper and lower optimality bounds $O\left(\sqrt{\frac{G^2D^2 + M^2m}{T}}\right)$ for GDRO. The upper bounds are achieved by using a two-player zero-sum game between the min and max players, where the min player is an online stochastic gradient descent algorithm while the max player is either online stochastic mirror descent (Algorithm 3) or EXP3.P (Algorithm 2). The information-theoretic lower bound construction is similar to that of bandits, where there are a number of different competing GDRO problem instances that differ slightly from a base environment. These problem instances induce similar sequence of oracle responses and reduce the probability that a learner could differentiate between the base and competing problem instances, leading to high error.

**Strengths:**

The paper presents a complete account of the optimal bounds for GDRO, showing both upper bounds and a matching lower bound. For the upper bound, while the main idea of using the two-player zero-sum game for optimization has been seen before (e.g. [1]), as the paper stated it is still a necessary and significant effort to put everything together to obtain a simple yet effective solution. The lower bound is significant since it is the first lower bound for GDRO. The bandit-nature of the construction shows new insights on the fundamental connection between bandits and GDRO, which I think is significant.

Another strength of the paper is the generalized GDRO setting, which make proposed approach and results applicable to other stochastic convex-concave problems of similar nature.

[1] Adaptive game playing using multiplicative weights. Freund and Schapire. 1999.

**Weaknesses:**

Theoretical results: while the theoretical results are interesting, they seem to be poorly presented. Their novelty is questionable as well. Details are below.
- Section 1 mentions that the most challenging part of the algorithm design is balancing the variance of the gradient estimators and the diameter. However, throughout the paper I could not find any part that explicitly deals with this variance. Does this "most challenging part" essentially boil down to tuning the learning rate $\eta_{\theta,t}$ based on $G, D, m$ and $t$?
- The results are reported in the expectation of the optimality gap, but the main paper never mentions what the expectation is over. It is in fact crucial to be specific early on what the expectation is over, because in the two-player zero-sum game, one player is an adaptive adversary to the other player, so the expected regret and pseudo-regret of the players have different interpretations.
- There are two different notions of optimality gap: $\epsilon_T$ and $\epsilon_T(q^*)$, however it was not discussed which one is more suitable and/or significant for GDRO. In particular, it is not clear which notion the lower bound in Theorem 5 applies to.
- The paper claims that by definition $\epsilon_T(q^*) \leq \epsilon_T$ for any $q^* \in Q$. This seems non-trivial and needs further clarification, because $\max_{q \in Q}L({\bar{\theta}}, q) \geq L({\bar{\theta}}, q)$ but $\max_{q \in Q}L(\theta^*, q) \geq L(\theta^*, q^*)$, so it is not immediate why $\max_{q \in Q}L({\bar{\theta}}, q) - \max_{q \in Q}L(\theta^*, q) \geq L({\bar{\theta}}, q^*) - L(\theta^*, q^*)$?
- Is the result of Theorem 2 really novel compared to the work of [2]? I know [2] uses EXP3-IX instead of EXP3.P, but they are just two different ways to get a high-probability regret bound for adversarial bandits.

[2] Stochastic Approximation Approaches to Group Distributionally Robust Optimization. Zhang et al. 2023.

Experiments:
- The number of groups in the majority of the experiments, including the one reported in the main paper, is less than 10. This is too small given that the point of the experiments was to show the improvement from $O(m\sqrt{\dots})$ to $O(\sqrt{m}\sqrt{\dots})$. Is there any inherent difficulty to run the experiments on a synthetic dataset with large $m$, say $m \in [1000, 5000]$?

- Even in the one experiment with $m = 100$ (Figure 2 in the appendix), at $T = 10^6$ the objective value of Sagawa et al. (the green line) is smaller than that of GDRO-INF, so I do not find the empirical evidence convincing. Can the authors comment on this?

Minor:
- Page 5: "In o" is not a sentence.
- Algorithm 1: the definition of the Bregman divergence is incorrect. The dot product should have $\nabla \Psi(y)$, not $\nabla \Psi(x)$.
- Theorem 1: What is $\mathbb{E}_{i_t}$? Again, this is the issue with unclear definition of the expectation.
- Throughout the proofs in the appendix, the paper makes use of existing results in Appendix A.1 that were stated for deterministic setting in which the actual gradients are available. However, in GDRO we only have the stochastic gradients, so it is important to be explicit on why the lemmas in A.1 were applied straightforwardly in the proofs.

**Questions:**

Please address the questions raised in the Weakness box above, especially on the expectation of the optimality gap, the mathematical correctness of writing $\epsilon_T(q^*) \leq \epsilon_T$ for any $q^* \in Q$, and the novelty of Theorem 2. I am willing to raise my score and vote for acceptance if those are addressed.

---

### Official Review · Reviewer_4T2i · 2023-10-30

**Soundness:** 2 fair
**Presentation:** 2 fair
**Contribution:** 3 good
**Rating:** 3
**Confidence:** 4

**Summary:**

This paper investigates group distributionally robust optimization (GDRO) problem. The authors propose propose two algorithms that can achieve near-optimal convergence rates, named GDRO-EXP3 and GDRO-TINF respectively. Furthermore, they also prove an almost matching information-theoretic lower bound for the convergence rate of GDRO.

**Strengths:**

This paper conducts thorough research on the GDRO problem, aiming to solve the DRO optimization problem in the general case.

**Weaknesses:**

See below.

**Questions:**

First, my confusion lies in the definition of the convergence rate. According to Sagawa et al. (2020), the convergence rate is formulated as,

\begin{equation}
\epsilon\_T = \max_{q\in Q} L(\bar{\theta}\_{1:T},q) - \min\_{\theta \in \Theta} \max\_{q \in Q} L(\theta,q)
\end{equation}

However, the authors adopt a different formulation,
$$
\epsilon\_T = \max\_{q\in Q} L(\bar{\theta}\_{1:T},q)-\max_{q\in Q} L(\theta^*,q).
$$
where $\theta^*$ is an optimal solution.
Is $\min_{\theta\in \Theta}\max_{q\in Q} L(\theta,q)=\max_{q\in Q} L(\theta^*,q)$?



 If they are equal, then in Appendix A.2, $\theta^*$ appears in regret,
$$
R_\theta (T;\theta^*)=\sum_{t=1}^T L(\theta_t,q_t) - \sum_{t=1}^T L(\theta^*,q_t).
$$
Are you sure that $\theta^*$ is the same solution in the definition of $\epsilon_T$ and $R_\theta (T;\theta^*)$ ? In my opinion, the optimal solution $\theta^*$ will change because it depends on $q$. Additionally, the authors define $\epsilon_T (q^*)$,
$$
\epsilon_T (q^*) = \max_{q\in Q} L(\bar{\theta}_{1:T},q)-L(\theta^*,q^*).
$$
where $q^*\in Q$ is any fixed point. Can you clarify the motivation that you define $\epsilon_T (q^*)$ ? I believe that $\epsilon_T (q^*)$ lacks significance because we cannot obtain the guarantee of $\epsilon_T$ from $\epsilon_T (q^*)$, i.e., $\epsilon_T (q^*)=O(\sqrt{m/T})\nRightarrow \epsilon_T =O(\sqrt{m/T})$. Since the authors redefine $\epsilon_T$ And $\epsilon_T (q^*)$, I believe comparing the convergence rate of the proposed algorithms  with Sagawa et al. (2020) in Table 1 is unfair and meaningless.


Second, there is an issue with the proof of Theorem 2. The authors attempt to utilize regret to bound the convergence rate. However, in online learning, the regret bound is formulated on random functions, i.e., $R(T)=\sum_{t=1}^T f_t(x_t)-\sum_{t=1}^T f_t(x_*)$. Therefore, the authors cannot directly use such bounds to achieve the expectation of convergence rate $\mathbb{E}[\epsilon_T]$in Theorem 2. If the authors modify the convergence rate in Theorem 2 to $\epsilon_T (q^*)$, then Theorem 2 can be directly substituted with online learning regret bound; otherwise, it cannot.

**Details Of Ethics Concerns:**

None.

---

### Official Review · Reviewer_tgUD · 2023-11-01

**Soundness:** 3 good
**Presentation:** 3 good
**Contribution:** 2 fair
**Rating:** 5
**Confidence:** 4

**Summary:**

This paper investigates a set of algorithms for solving DRO problems and the proposed method can obtain better convergence than existing methods. The authors also develop a lower bound showing that the proposed algorithm is optimal up to a log factor. Numerical experiments validate the effectiveness of the method proposed.

**Strengths:**

1. The authors propose two algorithms that achieve better convergence than existing methods.

2. This paper develops a lower bound for the problem, which shows the optimality of the proposed methods.

3. The methods are validated via numerical experiments.

**Weaknesses:**

1. The proposed algorithms seem simple and straightforward. It seems that the algorithm follows the structure of OMD and other existing online learning methods. It would be better if the authors could highlight the challenge and the novelty of the analyses.

2. The authors should discuss and compare more about other works of Group DRO, such as "On-demand sampling: Learning optimally from multiple distributions"

**Questions:**

See Weaknesses Part